# Estimation of the Minimum Effective Dose of Dietary Supplement Crocetin for Prevention of Myopia Progression in Mice

**DOI:** 10.3390/nu12010180

**Published:** 2020-01-09

**Authors:** Kiwako Mori, Toshihide Kurihara, Xiaoyan Jiang, Shin-ichi Ikeda, Erisa Yotsukura, Hidemasa Torii, Kazuo Tsubota

**Affiliations:** 1Department of Ophthalmology, Keio University School of Medicine, 35 Shinanomachi, Shinjuku-ku, Tokyo 160-8582, Japan; morikiwako@gmail.com (K.M.); jiaxiangya@yahoo.co.jp (X.J.); shin-ikeda@keio.jp (S.-i.I.); erisa.yotsuku@icloud.com (E.Y.); htorii@2004.jukuin.keio.ac.jp (H.T.); 2Laboratory of Photobiology, Keio University School of Medicine, 35 Shinanomachi, Shinjuku-ku, Tokyo 160-8582, Japan; 3Tsubota Laboratory, Inc., Keio University Shinanomachi Campus 2-5F, 35 Shinanomachi, Shinjuku-ku, Tokyo 160-8582, Japan

**Keywords:** crocetin, supplement, myopia, axial length, refraction, myopia control, minimum effective dose

## Abstract

The natural carotenoid crocetin has been reported to suppress phenotypes of an experimental myopia model in mice. We investigated the minimum effective dose to prevent myopia progression in a murine model. Three-week-old male mice (C57B6/J) were equipped with a −30 diopter (D) lens to induce myopia, and fed with normal chow, 0.0003%, or 0.001% of crocetin-containing chow. Changes in refractive errors and axial lengths (AL) were evaluated after three weeks. Pharmacokinetics of crocetin in the plasma and the eyeballs of mice was evaluated with specific high sensitivity quantitative analysis using liquid chromatography tandem mass spectrometry (LC-MS/MS) to determine the minimum effective dosage. A concentration of 0.001% of crocetin-containing chow showed a significant (*p* < 0.001) suppressive effect against both refractive and AL changes in the murine model. Meanwhile, there was no significant difference of AL change between the 0.0003% and the normal chow groups. The concentration of crocetin in the plasma and the eyeballs from mice fed with 0.001% crocetin-containing chow was significantly higher than control and 0.0003% crocetin-containing chow. In conclusion, we suggest 0.001% of crocetin-containing extract is the minimum effective dose showing a significant suppressive effect against both refractive and AL changes in the murine model.

## 1. Introduction

Myopia has been rapidly increasing in recent years worldwide [1]. Holden et al. suggested that myopia as well as high myopia would increase in the next 30 years estimating that the prevalence of myopia and high myopia would reach 5 billion and 1 billion, respectively, and that 40% of blindness would be attributed to myopia [1]. As the number of myopia increases, the number of high myopia and pathological myopia which may lead to myopic maculopathy, retinal detachment, and glaucoma is expected to increase [1]. The medical cost related to the increase of myopia will expand, and therefore, suppression of myopia is now more important [2]. Development of the way of myopia prevention is much more aggressively performed than before [3,4]. Although preventative measures such as outdoor activities, atropine eye drops, and orthokeratology lenses are under investigation [5], there have few oral medicines or supplements which showed significant suppressive effect on myopia progression.

The light environment was considered to be a significant factor for myopia progression [6,7,8]. Violet light is 360 to 400 nm wavelength in the visible light and shown to have a suppressive effect of myopia [9,10,11]. Furthermore, we reported that crocetin suppressed myopia progression in mice through the same mechanism as violet light exposure [12].

Crocetin was demonstrated to have a suppressive effect on myopia progression in mice, and the expression of *Egr-*1, a myopia suppressive gene, was shown to be in a dose-dependent manner in vitro [12]. In the previous study, crocetin-containing extract chow with two different concentrations, 0.003% and 0.03%, showed an equally suppressive effect in a murine model of lens-induced myopia [12]. Regarding lens-induced myopia, when equipped with minus diopter lenses, the axial length elongates to adjust the focus of the vision on the retina, which consequently makes the eye myopic. We previously revealed and reported that −30 D lenses are the most effective to produce lens-induced myopia phenotypes in mice after examining several lenses with different diopter, from −30 D to +5 D, to find which diopter is the most effective [13]. As it meant 0.003% of crocetin-containing extract was sufficient, the minimum required dose of crocetin to suppress myopia progression should be investigated for applying crocetin administration to human children and performing a randomized controlled trial. In the current study, we evaluated the animal model fed with either normal or two lower concentrations (0.001% and 0.0003%) of crocetin-containing extract chow.

## 2. Materials and Methods

### 2.1. Animals

All procedures were performed in accordance with the National Institutes of Health (NIH) guidelines for work with laboratory animals and the Association for Research in Vision and Ophthalmology (ARVO) statement for the Use of Animals in Ophthalmic and Vision Research, and were approved by the Institutional Animal Care and Use Committee at Keio University (registration no. 3930). C57BL6/J mice (CLEA Japan, Tokyo, Japan) were raised in standard transparent mouse cages (29 × 18 × 13 cm) in an air-conditioned room maintained at 23 ± 3 °C under a 12 h dark/light cycle, with free access to a diet (MF, Oriental Yeast Co., Ltd., Tokyo, Japan) and tap water. Four or five mice were kept in one cage with 12 h on/off [13].

Our study was also in compliance with Animal Research: Reporting of In Vivo Experiments (ARRIVE) guidelines. Assignment of the animals to each group was as follows; each animal was randomly assigned to one of the groups, control group, 0.0003% (1 mg/kg/day) crocetin-containing extract group, and 0.001% (3 mg/kg/day) crocetin-containing extract group in a completely masked manner. The mice were numbered to hide the treatment status. The researchers who bled the mice and ones who measured the refractive error were different. Therefore, the researchers were completely masked to the treatment status of animals during measurements of refractive errors. Due to the variability, mice with only male mice of the same body weight (10 ± 1 g) were prepared.

### 2.2. In Vivo Analysis with the Dietary Factor in the Murine Model of Lens-Induced Myopia

For the in vivo analysis, we performed the experiment of a lens-induced myopia in mice as previously reported [13]. Briefly, we designed a frame of eyeglasses for mice accommodating to the shape of the mouse head and outputted it using a three-dimensional printer. For myopia induction, a negative 30 diopter (D) lens made of poly methyl methacrylate (PMMA) was designed and the side of the frame was fit to equip the right eye with the lens. A 0 D lens was fixed to the left eye as an internal control. Three-week-old wild-type C57BL/6J male mice were equipped with the eyeglasses by mounting the frame on their skull using a self-cure dental adhesive system under general anesthesia with the combination of midazolam (Sandoz K.K., Tokyo, Japan), medetomidine (Domitor^®^, Orion Corporation, Espoo, Finland) and butorphanol tartrate (Meiji Seika Pharma Co., Ltd., Tokyo, Japan) (MMB). A 0.01mL/g of MMB was administered intraperitoneally. Animals were fed with normal (MF, Oriental Yeast Co., Ltd., Tokyo, Japan) or mixed chow containing 0.0003% (1 mg/kg/day) and 0.001% (3 mg/kg/day) of crocetin-containing extract (Crovit P, compound containing more than 75% of crocetin, RIKEN VITAMIN Co., Ltd., Tokyo, Japan) during the period of the myopia induction. The process of addition of crocetin-containing extract to the chow in this study was managed by the manufacturer. (Oriental Yeast Co., LTD, Tokyo, Japan) [12].

### 2.3. Ocular Components Measurement

We measured the ocular components in mice as previously described [12,13]. Refraction and AL were measured at the initial (3 week-old) and the end (6 week-old) stage of the myopic induction using an infrared photorefractor (Steinbeis Transfer Center, Germany) and a spectral domain optical coherence tomography (SD-OCT) system (Envisu R4310, Leica, Wetzlar, Germany). All measurements were performed under mydriasis by 0.5% tropicamide and 0.5% phenylephrine eye drops (Santen, Osaka, Japan), and general anesthesia by MMB. The refractive error values were averaged with 100 times of the measurement in a continuous data trace. The AL was determined from the anterior corneal surface to the retinal pigment epithelium along the corneal vertex reflection. The corneal curvature radius was measured by an infrared keratometer (Steinbeis Transfer Center, Tübingen, Germany) [13].

### 2.4. Crocetin Concentration in the Eyeballs and Plasma

Pharmacokinetics of crocetin in the plasma and eyeballs of the lens-induced myopia mice was evaluated with specific high sensitivity quantitative analysis using liquid chromatography tandem mass spectrometry (LC-MS/MS).

#### 2.4.1. Sample Preparation

The samples were prepared as follows; 9 mice fed with control chow, 8 mice fed with 0.0003% crocetin-containing extract chow, and 10 mice fed with 0.001% crocetin-containing extract chow. Mice were anesthetized with overdosed MMB three weeks after fed with normal or crocetin-containing chow. Blood was collected from the inferior vena cava using a 24-gauge needle and a 1 mL syringe which was filled with heparin sodium and dried up. The whole blood was transferred into 1.5 mL tubes (Eppendorf, Hamburg, Germany) and they were centrifuged at 10,000× rpm for 3 min at 4 °C followed by collecting the plasma. After collecting the samples, enucleation of right and left eyes in order was performed, adhering adipose tissues were removed, and the eyes were frozen with liquid nitrogen.

#### 2.4.2. Chemicals and Reagents

Crocetin reference standard (95% pure) was obtained from Toronto Research Chemicals (Canada). Niflumic acid (Merck, Darmstadt, Germany) was employed as an internal standard (IS). LC/MS grade acetonitrile and methanol were purchased from Kanto Chemical Co., Inc. (Japan). HPLC grade ammonium acetate was purchased from Fujifilm Wako Pure Chemical Corporation (Japan). Dimethyl sulfoxide (DMSO) was purchased from Nacalai Tesque (Japan). Water used for the LC-MS/MS analysis was purified by Auto Pure WR600G (Yamato Scientific Co., Ltd., Tokyo, Japan).

#### 2.4.3. Preparation of Calibration Standards and Quality Control (QC) Samples

The stock standard solution of crocetin (1 mg/mL) was prepared by dissolving in DMSO. The working standard solutions (0.1 to 10 ng/mL) were prepared by diluting in acetonitrile. Calibration standards and QC samples were prepared by spiking working standard solutions and IS into blank mouse plasma or 20% eyeballs homogenates (Charles River Laboratories Japan, Inc., Kanagawa, Japan). The final concentrations of calibration standard samples were 1–100 ng/mL for the plasma and 0.1–10 ng/g for the eyeball. QC samples were prepared at the concentrations 3, 10, and 80 ng/mL for the plasma and 0.3, 1, and 8 ng/g for the eyeballs.

#### 2.4.4. Sample Preparation Procedure

To make a 10 µL plasma sample or 50 μL of the above eyeball homogenates, 50 μL of IS (1 ng/mL acetonitrile solution) and 10 µL of acetonitrile were added to precipitate the proteins in the sample. The resulting solution was transferred to a HPLC vial and 160 µL of mobile phase A was added. The mixture was injected into LC-MS/MS.

#### 2.4.5. LC-MS/MS Conditions

LC-MS/MS analyses were performed using a high-performance liquid chromatography (HPLC) system (ACQUITY UPLC I-Class Systems (Waters, Milford, MA, USA)) and Xevo TQ-XS (Waters, Milford, MA, USA). Chromatography separation was performed using an InertSustain C18 column (5 µm, 2.1 mm I.D. × 50 mm, GL Sciences, Tokyo, Japan). MS/MS analyses were conducted in negative-ion mode, and crocetin were identified and quantified by multiple reaction monitoring (MRM) (Appendix A).

### 2.5. Statistical Analyses

All results in animal experiments were expressed as mean ± standard deviation (SD). A t-test or post-hoc Tukey test with one-way analysis of variance (ANOVA) was used to assess the statistical significance of the differences (Microsoft Excel 2013, Redmond, WA, USA and IBM SPSS statistics version 23, Chicago, IL, USA), and results with *p*-values < 0.05 were considered significant.

## 3. Results

### 3.1. Determination of Minimum Effective Dosage with a Murine Model of Lens-Induced Myopia

Normal or two different concentrations, 0.001% and 0.0003%, of crocetin-containing extract chow was evaluated. Eyes treated with −30 D lenses showed a significant (*p* < 0.001) myopic change (−14.46 ± 2.11 D) compared to ones with 0 D (+8.87 ± 2.63 D) in the normal chow-fed animals (the control group) (Figure 1a). Animals fed with crocetin-containing chow showed a significantly smaller refractive change with −30 D lens (+2.46 ± 2.22 D for 0.001%, −1.28 ± 3.62 D for 0.0003%) compared to the control group with −30 D (−14.46 ± 2.11 D) (*p* < 0.001 for both concentrations, Figure 1a).

The change of AL in the eyes with −30 D lenses in the control group showed a significantly (*p* < 0.01) larger (0.25 ± 0.03 mm) compared to ones with 0 D (0.20 ± 0.03 mm) (Figure 1b). The change of AL in the eyes with −30 D lenses in the 0.001% crocetin-containing extract chow group showed a significantly (*p* < 0.001) smaller (0.19 ± 0.02 mm) compared to those in the eyes with −30 D lenses (0.25 ± 0.03 mm) in the control group (Figure 1b). In contrast, the 0.0003% crocetin-containing extract chow group showed no significant change of AL in the eyes with −30 D lenses compared to the control group with −30 D lenses (0.22 ± 0.03 mm for the 0.0003%, 0.25 ± 0.03 mm for the control) (*p* = 0.136, Figure 1b).

The 0.001% crocetin-containing extract chow group showed suppression of both refractive and AL change in the murine myopia model while there was no significant difference of AL change between the 0.0003% and the control groups indicating that 0.001% is the minimum effective dosage with the model.

### 3.2. Crocetin Concentrations in the Plasma and Eyeballs in Mice

Concentrations of crocetin in the plasma and the eyeballs of mice fed with each chow were measured with liquid chromatography tandem mass spectrometry. The concentration of crocetin in the plasma of mice fed with 0.001% crocetin-containing extract chow was significantly (*p* = 0.043) higher than that of mice fed with 0.0003% crocetin-containing extract chow. Crocetin was not detected in the plasma of mice fed with control chow (Table 1, Figure 2a). The concentration of crocetin in the eyeballs with mice fed with 0.001% crocetin-containing extract chow was significantly (*p* < 0.001) higher than that of mice fed with control and 0.0003% crocetin-containing extract chow (Table 1, Figure 2b).

## 4. Discussion

Previous studies demonstrated that both of the concentrations, 0.003% and 0.03%, of crocetin-containing extract equally have a suppressive effect on myopia progression [12]. Due to this result, we investigated the minimum required dose to suppress myopia progression in mice. As a result, 0.0003% of crocetin-containing extract showed a suppressive effect on myopic refractive change but it showed no suppressive effect on axial elongation, whereas 0.001% of crocetin-containing extract showed an effect on both refractive change and axial elongation. This result gave us the idea that the critical point existed between these concentrations.

The concentration of crocetin detected in the eyeballs with 0.0003% crocetin-containing extract chow was not different from that with control chow. Meanwhile, when 0.001% crocetin-containing chow was administered, the concentration in the eye balls was significantly different from those with 0.003% crocetin-containing extract chow as well as the control chow, which corresponded to phenotypes of refraction and AL and supported the myopia suppressive effect of 0.001% crocetin-containing extract chow.

We prepared the concentrations of crocetin-containing chows so that 3 mg/kg/day of Crovit P were administered when a three-week-old B6 mouse consumes 0.001% of crocetin-containing extract per day. Crovit P used in this study is a product which guarantees more than 75% purity of crocetin. In the meantime, the body surface area of a 10 g body weight mouse is 0.0035 m^2^, whereas the body surface area of a 10 year-old child, who is thought to be predisposed to myopia progression, is generally 1.15 m^2^. We estimated the body surface area of an average 10 year-old child using the Du Bois and Du Bois formula based on the statistical survey of school health in 2018 in Japan. Consequently, 0.0001% crocetin-containing extract chow corresponds to 10 mg/day of Crovit P, which exceeds 7.5 mg/day of pure crocetin.

When excerpted from previous human studies on crocetin administration, the dose varies between 7.5 mg to 22.5 mg [14]. 0.0001% crocetin-containing extract chow, the minimum required dose to suppress myopia, corresponds to 7.5 mg/day when administered to human. Umigai et al. reported that 7.5 mg to 22.5 mg of crocetin can be safely administered to human without any side effects [14].

The question whether the age of mice and the period of administration of crocetin are proper can be raised. Regarding the age, mice at three weeks of age, which corresponds to the adolescent period in humans, were most susceptible to experimental myopia induction [13]. In addition, the period of three weeks to induce myopia by applying −30 diopter lenses in mice is found to be good enough to obtain statistical significant differences [13]. As for the timing of the initiation of crocetin administration to human children, it is presumed that the treatment should be started by their adolescence.

The difference in AL between the control group and 0.001% crocetin-containing chow group was 0.06 mm with a statistical significance. The mean murine ocular axial length is approximately 3.264 ± 0.047 mm measured by a SD-OCT [15]. The human full-term newborn eye has a mean axial length of 16–18 mm [16]. The mean human adult values for axial length are 22–25 mm and mean refractive power −25.0 ± 1.0 D [16]. The difference in AL of 0.06 mm in mice corresponds to approximately 0.43 mm in humans, which can be considered clinically significant.

In this study, crocetin was detected in the eyes of mice fed with crocetin in a dose-dependent manner, whereas a small amount of crocetin was also found in the eyes of control mice fed with normal chows which contained no crocetin. To exclude the possibility of contamination, crocetin in the eyes of mice bred in other facility which is free from crocetin was measured; consequently, crocetin was also detected in the eyes of these mice. The crocetin detected in the eyes of control mice were thought to be derived from the pigments which naturally exists in the retina. Bowmaker et al. previously reported that pigments which was considered to be crocetin existed in the retina [17]. There have been few reports regarding pharmacodynamics of crocetin in the eyes, and the efficacy and the role of crocetin have not sufficiently been proven.

This study has some limitations. Though extracts of crocetin are mixed to chows in this research, it is technically difficult to prepare low concentrations, less than 0.0003%, of crocetin-containing extract chow. This should be further investigated.

This research was fundamentally aimed at application of administration of crocetin to humans. Myopia has been increasing worldwide and is anticipated to continue to increase. Possible preventative measures against myopia progression and treatments for myopia are now eagerly under investigation. Crocetin can be one of the hopeful candidates as a myopia preventive method and expected to be applied to humans. We hope our study will help future investigation of myopia prevention.

## 5. Patents

Patents have been registered for the therapeutic effects of crocetin (patent no. 6,502,603 by Tsubota Laboratory, Inc., Tokyo, Japan, and Rohto Pharmaceutical Co., Ltd., Osaka, Japan.) and applied for the design of the mouse eyeglass by Tsubota Laboratory, Inc. (patent application no. 2017-41349).

## Figures and Tables

**Figure 1 nutrients-12-00180-f001:**
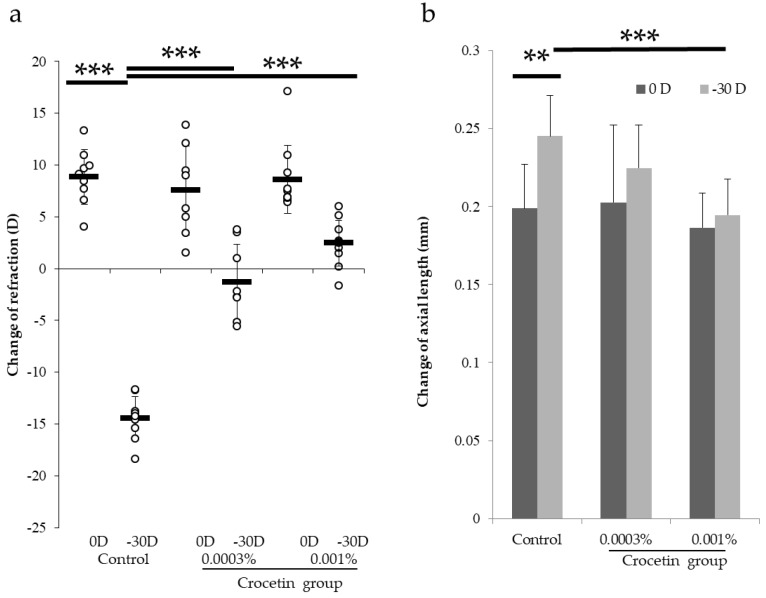
0.001% crocetin-containing extract chow suppressed myopic progression in the murine model of lens-induced myopia. (**a**) Animals fed with 0.0003% and 0.001% crocetin-containing extract chow showed a significantly (*p* < 0.001) smaller refractive change with −30 D lens compared to the same condition in the control. (**b**) The change of AL in the eyes with −30 D lenses showed a significantly (*p* < 0.01) larger compared to those in the eyes with 0 D in the control group. The change of AL in the eyes with −30 D lenses in the 0.001% crocetin-containing extract chow group showed a significantly (*p* < 0.001) smaller compared to those in the eyes with −30 D lenses in the control group. There were no significant differences in the changes of AL in both eyes between the control group and the 0.0003% crocetin-containing extract chow group. ** *p* < 0.01, *** *p* < 0.001. Bars represent mean ± standard deviations.

**Figure 2 nutrients-12-00180-f002:**
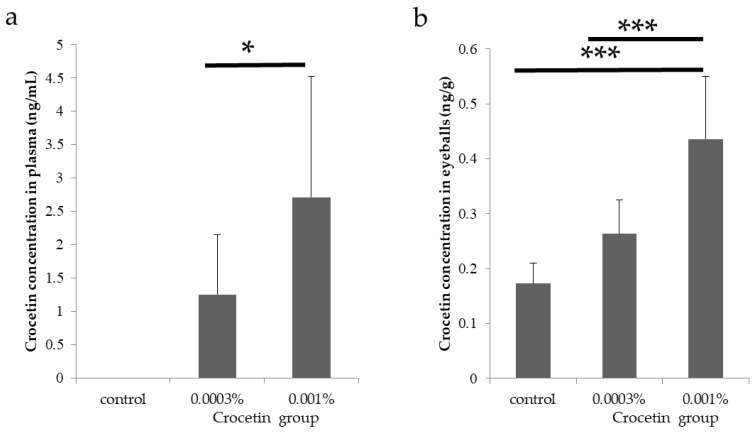
Crocetin concentrations in the plasma and eyeballs in mice. Concentrations of crocetin in the plasma and the eyes of mice fed with each chow were measured with liquid chromatography tandem mass spectrometry. (**a**) Concentration of crocetin in the plasma of mice fed with 0.001% crocetin-containing extract chow was significantly (*p* = 0.043) higher than that of mice fed with 0.0003% crocetin-containing extract chow. Crocetin was not detected in the plasma of mice fed with control chow. (**b**) Concentration of crocetin was significantly (*p* < 0.001) higher in the eyeballs of mice fed with 0.001% crocetin-containing extract chow than those in other groups. * *p* < 0.05, *** *p* < 0.001. Bars represent mean ± standard deviations.

**Table 1 nutrients-12-00180-t001:** Crocetin concentrations in the plasma and eyeballs in mice.

Type of Food	Concentration of Crocetin							
	Plasma (ng/mL)			*p* Value	Eyeballs (ng/g)			*p* Value
Control	NC				0.17	±	0.04	
0.0003% crocetin	1.25	±	0.89		0.26	±	0.06	
0.001% crocetin	2.71	±	1.81		0.44	±	0.11	
				0.043 ^†^				<0.001 ^††^

Data represent means ± standard deviations (SD), NC: Not calculated, ^†^: *t*-test, ^††^: Analysis of variance (ANOVA).

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
