# Peer review of "Estimation of the Minimum Effective Dose of Dietary Supplement Crocetin for Prevention of Myopia Progression in Mice"

_nutrients, 2020, doi:10.3390/nu12010180_

Round 1
Reviewer 1 Report
Overall: The paper appears to be the report of a related study to those of many of the same authors to two previous studies (published Jan and Jun 2019) that examined the effects of myopic shifts/axial length changes with crocetin administration in mice and children, respectively (see DOI:10.1038/s41598-018-36576-w & J. Clin. Med. 2019, 8, 1179; doi:10.3390/jcm8081179). The current investigation is novel in that it suggests a minimum % concentration of crocetin-containing chow or extract. The manuscript is very well-written, although I have made many minor suggestions for clarity. The authors have also convinced me that their data are reliable and methods are sound. My biggest (and only) complaint is in the strength of their conclusion. I suggest the authors report "We suggest that the minimum % effective is...." rather than stating that they have found the answer fully. If the manuscript is published, and I write in favor of that outcome, I have made many minor suggestions throughout.
Line 17: Perhaps, "We investigated..." instead
Line 20: Suggest "Changes in..." (instead of "changes of..."
Line 23: It is awkward to start a sentence with a number like this (0.00...). Suggest re-arranging the sentence.
Line 28: Suggest, "In conclusion, we suggest 0.001% of crocetin-containing extract as the minimum dose..."
Line 74: First line needs indented.
Line 77 (and throughout manuscript): Suggest "...crocetin-containing..."
Line 81: Suggest "...., only male mice of same body weight..."
Line 96-97: This last sentence is a little awkward.
Line 99: Suggest: "We measured the ocular components as previously described [12]."
Line 111: Suggest "with specific high sensitivity quantitative..." (i.e., remove "establishment of")
Line 112: Need end parenthesis.
Line 134: need space before "ng/g"
Lines 208-209: I suggest "...a suppressive effect on myopia progression..." (i.e. remove "...a suppressive effect on refractive change leading to..."
Lines 216 & 217: I suggest removing everything after the first "chow" on Line 216.
Line 221: Please add space between 0.0035 and m2
Line 227: Please change "...researches of crocetin administration to humans," to "...from previous human research on crocetin administration..."
Lines 235-236: Suggest changing "; as a result, crocetin was also detected." to "; crocetin was also detected in the eyes of these mice."
Lines 245-246: Suggest removing "The minimum effective dose of crocetin in humans was calculated by that in mice using the body surface area method which cannot be applied universally but can be used as an index."
Reviewer 2 Report
The first question is that , as described, a -30D lens will induce hyperopia rather than myopia. To induce myopia, it would require a +30D lens.
I think there needs to be some justification of the time period (3weeks) of the study. It begs the question, is that long enough to show significant differences. Also, what age might this correspond to in humans?
In results and discussion - although a difference in AL of .06 mm was statistically significant, is it clinically significant?
In addition to calculated dosage for humans, at what age might this treatment be initiated ?
Specific comments
line 41, change researches to "studies"
line 77, used masked rather than blind
Line 155, delete the word "either" and reword Normal and ... were
line 206, change to "due to this result.."
line 227, change researches to "studies"
